# Nanometer-precision non-local deformation reconstruction using nanodiamond sensing

Kangwei Xia[1,3], Chu-Feng Liu[1,3], Weng-Hang Leong [1,3], Man-Hin Kwok[1], Zhi-Yuan Yang[1], Xi Feng[1], Ren-Bao Liu [1,2] & Quan Li[1,2]

Spatially resolved information about material deformation upon loading is critical to evaluating mechanical properties of materials, and to understanding mechano-response of live systems. Existing techniques may access local properties of materials at nanoscale, but not at locations away from the force-loading positions. Moreover, interpretation of the local measurement relies on correct modeling, the validation of which is not straightforward. Here we demonstrate an approach to evaluating non-local material deformation based on the integration of nanodiamond orientation sensing and atomic force microscopy nanoindentation. This approach features a 5 nm precision in the loading direction and a sub-hundred nanometer lateral resolution, high enough to disclose the surface/interface effects in the material deformation. The non-local deformation profile can validate the models needed for mechanical property determination. The non-local nanometer-precision sensing of deformation facilitates studying mechanical response of complex material systems ranging from impact transfer in nanocomposites to mechano-response of live systems.

[1] Department of Physics, The Chinese University of Hong Kong, Shatin, New Territories, Hong Kong, China. [2] The Hong Kong Institute of Quantum Information Science and Technology, The Chinese University of Hong Kong, Shatin, New Territories, Hong Kong, China. [3] These authors contributed equally: Kangwei Xia, Chu-Feng Liu, Weng-Hang Leong. Correspondence and requests for materials should be addressed to R.-B.L. (email: rbliu@cuhk.edu.hk) or to Q.L. (email: liquan@phy.cuhk.edu.hk)

Deformation measurements of heterogeneous nano-materials[1,2] (e.g., nanocomposites and multiphase poly-mers) and live systems[3–5] (e.g., microbes and cells) are critical in evaluating their mechanical properties, and more importantly, provide a viable means to study the mechano-stimulation response of live systems[6,7]. Nanoindentation[8], a method that gives the loading depth profile of materials, is commonly used to investigate the mechanical response upon loading[9,10]. Although local deformation at the contact region can be obtained by well-established methods[11–14], the resolution of the measurement remains unsatisfactory. More importantly, the measured local deformation results from a summation of sample deformation and cantilever deflection. It largely depends on the conditions of the indentation, such as the type of indenter, the indentation speed, the type of liquid employed, the humidity, etc.[15,16]. If soft matter or biological systems are studied, the viscosity of the samples also plays a role. Using only local deformation profiles is usually inadequate to determine the material mechanical properties. On the other hand, the estab-lished methods miss the information about the non-local response of the material to the local loading; such information is essential to understanding the deformation and impact transfer in heterogeneous structures, and how live systems respond to mechano-stimuli. Up to date, the only deformation measurement method that can extract non-local information is based on optical imaging[17–22]. The precision and the resolution of the optical measurement, however, is limited by the optical wavelength; or even with the super-resolution techniques based on single-molecule fluorescence, they are still limited to tens of nanometers. In typical optical measurement of indentation-induced defor-mation, indenters of large tip size (a few micrometers) are usually used to generate deformation large enough to be measurable (>a few micrometers)[19–22]. Such techniques lack the sensitivity required to probe important deformation phenomena, such as the effects of surface tension at the interfaces between soft materials and liquids. This is particularly relevant to the mechanical property measurement of soft matters, such as single cells, when accurate determination of the stiffness cannot be done without knowledge of the elastocapillary effect[19–23]. Therefore, it is highly desirable to develop a new method that can have both non-local access of deformation and nanometer precision. Such a tool would not only enable the investigation of impact transfer at the nanoscale in heterostructured materials with multiple phases and interfaces, but also facilitate the study of fine deformation char-acteristics of soft materials under weak external perturbation.

A new opportunity is diamond sensing[24–26]. Diamond sensing uses nitrogen-vacancy (NV) center[27] spins as the sensor, which can work under ambient conditions[28–32]. NV center spin reso-nances are sensitive to small changes of the magnetic field, making them effective quantum sensors to many physical para-meters via direct or indirect magnetometry measurements[33–36]. In particular, since the spin resonance frequencies are shifted mostly by the magnetic field component projected to the nitrogen-vacancy axis, vector magnetometry has been developed to determine the relative orientation between the magnetic field and the crystallographic direction of the diamond lattice[30,32,37–39]. The deformation is related to the orientation of the surface and hence the rotation of the nanodiamonds (NDs) that are situated on the surface. Based on this, with the combination of atomic force microscopy (AFM)-induced indentation, we propose and demonstrate a scheme of non-local deformation sensing using the rotation measurement of NDs on a surface.

We reconstruct the non-local surface topography using the three-dimensional rotation data of NDs docked on surfaces that are deformed by AFM indentation. We adopt two data-acquisition approaches—one with an ND fixed on a position and the AFM indentation position scans around the ND (single-ND approach); another with a fixed indentation position and multiple NDs locates at various positions on the surface (multi-ND approach). In the proof-of-the-concept demonstrations, we employ the single-ND approach and reconstruct the deformation of a polydimethylsiloxane (PDMS) film with a 5-nm precision in the loading direction and an in-plane spatial resolution ~200 nm (as determined by the AFM parameters—the scanning step size in the present experiments). The measured mechanical property of the PDMS film is well described by a linear elastic model that has been established in the literature[40]; this good agreement validates the method. We further apply this sensing method to measure deformation of gelatin microgel particles in water, using the multi-ND approach. We measure the effect of surface tension at the gelatin/water interface under nanoscale indentation. The load-responsive sensing protocols based on the ND rotation measurement provide an approach to studying spatially resolved non-local deformation, which features a precision <10 nm, bringing opportunities to study mechanical properties of non-uniform media and paving the way for investigating mechanical properties/mechano-stimuli-induced response of live systems in the future.

## Results

**Deformation reconstruction from ND rotation data**. Figure 1a illustrates the measurement scheme. We consider the deforma-tion of a material surface induced by a localized nanoindentation with an AFM tip. NDs are distributed across the material surface. The deformation would cause the rotation of the NDs. The rotation of the NDs is measured by optically detected magnetic resonance (ODMR) of the NV centers in the NDs under external magnetic fields. The nonlocal deformation of the surface (at positions other than the indentation location) is thus recon-structed using the ND rotation data.

The NV centers in an ND have four inequivalent crystal-lographic directions[27] ($NV_i$, $i = 1,2,3,4$, along $(111)$, $(\bar{1}\bar{1}1)$, $(1\bar{1}\bar{1})$, and $(\bar{1}1\bar{1})$ correspondingly) as shown in Fig. 1b. The ground state of the NV center is a spin triplet ($S = 1$) with a zero-field splitting $D \approx 2.87$ GHz between the $m_S = 0$ and $m_S = \pm 1$ states, which are quantized along the NV axis. In the presence of a weak magnetic field $\mathbf{B}$ (<100 Gauss), the frequencies of the transitions $|0\rangle \leftrightarrow |\pm\rangle$ are $f_i^{\pm} \approx D \pm \gamma_e B \cos \psi_i$, where $\psi_i$ is the angle between the NV axis ($NV_i$) and the magnetic field $\mathbf{B}$ (Fig. 1b), and $\gamma_e \approx 28$ MHz mT$^{-1}$ is the electron gyromagnetic ratio[25]. The angles $\psi_i$ and in turn the orientations of the NV axes are determined from the transition frequencies measured by the ODMR spectroscopy. Note that the four angles $\psi_i$ of the NV centers and hence the ODMR frequencies would be identical if the ND is rotated about the magnetic field by any angle. To remove such ambiguity in determination of the orientation of the ND, we applied a second magnetic field $\mathbf{B}'$ so that the angles $\psi_i'$ between the NV axes and $\mathbf{B}'$ are determined from the new ODMR spectra (Fig. 1c). This way, the three-dimensional orientation of the ND is unambiguously determined. The rotation of the ND is characterized by the rotation axis ($\theta_r, \phi_r$), with $\theta_r$ and $\phi_r$ being the polar and azimuthal angles, and the rotation angle $\chi$ (Fig. 1d). The rotation axis and angle ($\theta_r, \phi_r, \chi$) of the ND were deduced from the ODMR spectra before and after the rotation (details in the "Methods" section and Supplementary Note 2). The rotation measurement protocol was first validated using a bulk diamond (with an ensemble of NV centers) of known crystallographic orientation and rotation operations (see Supplementary Note 3).

The deformation was reconstructed from the rotation of the NDs docked on the surface in the following procedure. The rotation of an ND is characterized by the rotation matrix

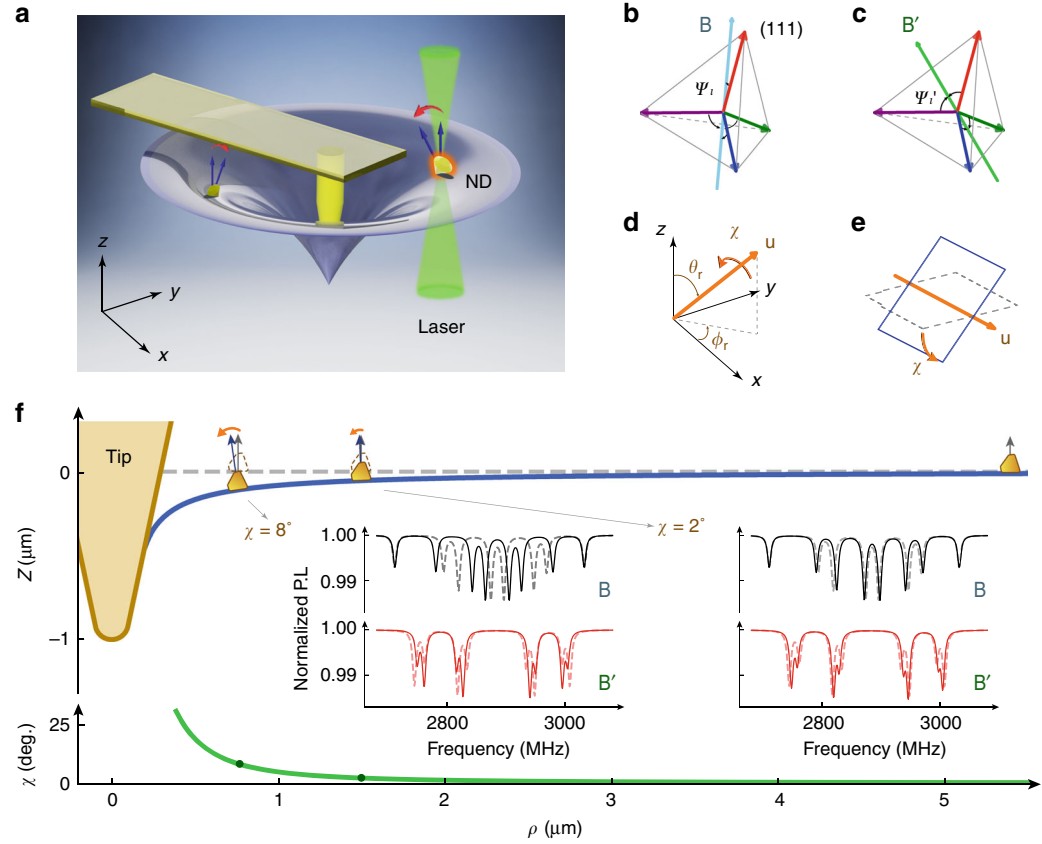

**Fig. 1** Method of deformation reconstruction via the rotation of nanodiamonds (NDs). **a** An atomic force microscopy (AFM) tip imposes a localized indentation, causing non-local deformation of the surface. The NDs docked on the surface are rotated due to the deformation. **b** and **c** Nitrogen-vacancy (NV) centers of four different orientations, with angles $\psi_i$ and $\psi_i'$ ($i = 1,2,3,4$) from two external magnetic fields **B** and **B′**, respectively. **d** Rotation of the ND is characterized by the direction of the rotation axis ($\theta_r,\phi_r$) and the rotation angle $\chi$. **e** The tangential surfaces before (dotted) and after (solid) the deformation, in which the intersection and the angle between the two surfaces correspond respectively to the rotation axis and the rotation angle of the ND docked at the surface. **f** Simulated deformation (blue line) upon an AFM tip indentation on a homogeneous material. The green line shows the simulated rotation angles of NDs as functions of their distance from the AFM tip. Inset: Examples of the simulated optically detected magnetic resonance (ODMR) spectra of the two NDs on the surface (indicated by arrows) under magnetic field **B** (black) or **B′** (red) with (solid line) or without (dash line) the indentation

$\mathbf{R}(\theta_r,\phi_r,\chi)$. The position of the ND is denoted as $\boldsymbol{\rho} = (\rho,\theta)$, written in the polar coordinates. The relation between the rotation of the ND and the rotation of the tangential plane of the surface at $\rho$, as illustrated in Fig. 1e, is given by the rotation of the normal direction, $\hat{\mathbf{n}}(\boldsymbol{\rho}) = \mathbf{R}(\theta_r,\phi_r,\chi)\hat{\mathbf{z}}$, where $\hat{\mathbf{z}}$ is the normal direction of the surface before the indentation (the z-axis). The displacement of the surface at the position of the ND, denoted as $z(\boldsymbol{\rho})$, was reconstructed by the least-square fitting of the rotation data, using the gradient field equation $\mathbf{g}(\boldsymbol{\rho}) = \nabla_\rho z(\boldsymbol{\rho})$[41], where the gradient is related to the rotated normal direction by $\mathbf{n} = (-g_x, -g_y, 1)$ (for details see Methods). In reconstructing the deformation, we assumed that the attached NDs had no relative motion or rotation against the surface. This assumption is consistent with the fact that the NDs recovered their original orientation after the indentation was released (see Supplementary Fig. 11).

To check the feasibility of the method, we ran numerical simulations of the ODMR spectra of NDs docked on the surface of a soft material under an AFM indentation (see Fig. 1f). We considered a homogeneous material upon an AFM tip indentation (Fig. 1f) and adopted the linear elastic model (the Hertz–Sneddon model)[42,43]. In the simulations, the Young's modulus $E = 30$ kPa and the load force $P = 10$ nN were used. The deformed surface and the rotation angle of the normal direction ($\chi$) are shown in blue and green lines in Fig. 1f, respectively.

Taking two NDs at different distances from the AFM tip as examples, we simulated their ODMR spectra under **B** or **B′** with or without the indentation (see insets in Fig. 1f). In both NDs, eight ODMR resonance dips were found, corresponding to the four pairs of $+/-$ transition frequencies of the NV centers along the four crystallographic orientations. The frequency shifts were clearly seen between the spectra with (solid line) and those without (dash line) the indentation; from the frequency shifts, the rotation of the NDs was deduced.

**Non-local deformation reconstruction—single-ND method.** For a laterally homogeneous material, the deformation upon a nanoindentation can be reconstructed with an ND sensor fixed on a position and an AFM tip scanning around the ND. This single-ND method is possible because on the homogeneous material, the deformation at position A upon an indentation at position B should be the same as the deformation at B upon an indentation at A.

We demonstrated the single-ND method using a PDMS film. The thickness of the PDMS film was ~50 μm with an oxidized surface layer formed by $O_2$ plasma treatment (for the preparation and the characterization of the PDMS film see Methods and Supplementary Note 4). NDs with a typical size of ~150 nm were dispersed on the surface of the PDMS film by spin coating.

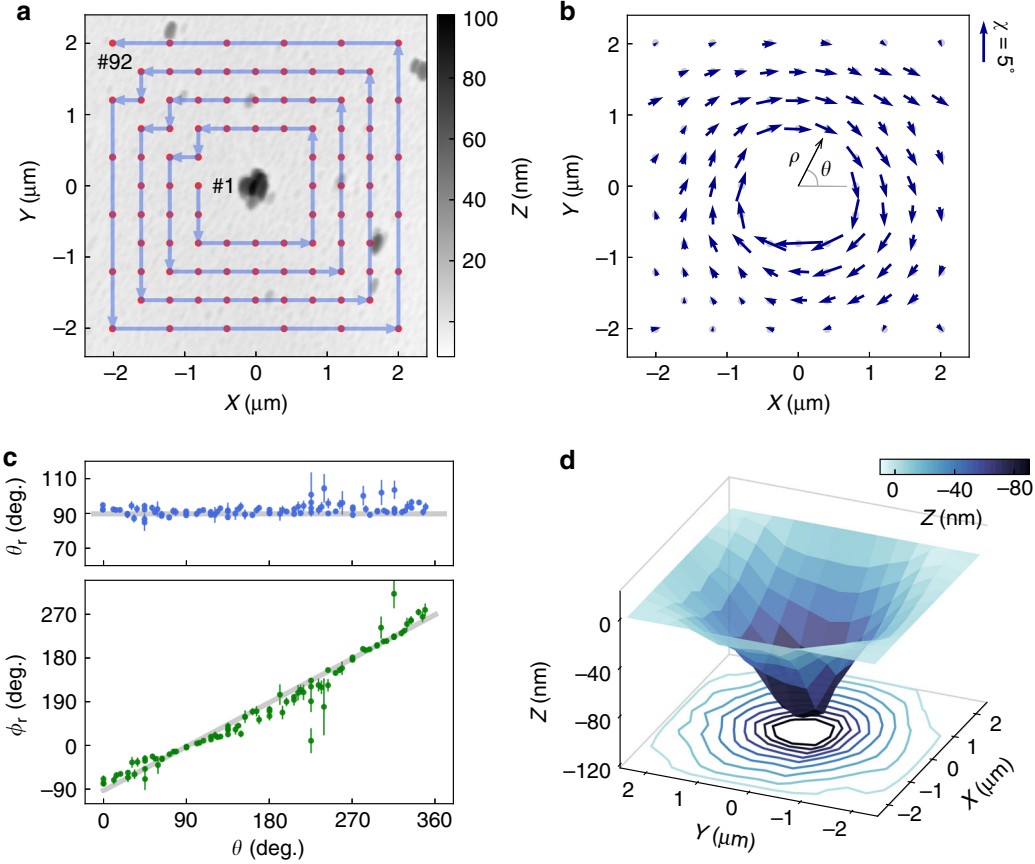

**Fig. 2** Deformation reconstruction using the single-nanodiamond (ND) method. **a** Atomic force microscopy (AFM) image of a typical polydimethylsiloxane (PDMS) surface with an ND located at the center (the origin). The red dots represent the indentation locations of the AFM tip, starting with #1 and ending with #92. **b** The rotation of the ND plot as arrows for the displacements of the ND from the 92 indentation positions. The size of the arrows represents the magnitude of the rotation angles $\chi$ (scale bar shown on the right), and the direction is the rotation axes projected to the $x$–$y$ plane. **c** The polar angle $\theta_r$ and the azimuthal angle $\phi_r$ of the rotation axes of the ND as functions of $\theta$ for the 92 indentation positions. The gray line in the upper graph is $\theta_r = 90°$ and in the lower graph is $\phi_r = \theta - 90°$. **d** The reconstructed surface of the PDMS film upon an AFM tip indentation at the center (the origin). The vertical error bars in **c** are the fitting errors

Figure 2a shows an AFM image of the PDMS surface with a single ND located in the center (0,0). The homogeneity of the PDMS film was confirmed by the almost identical load-depth profiles obtained at different locations on the sample by AFM indentation (see Supplementary Fig. 5b). The comparison between the AFM image and the confocal image taken from the same area helped to determine the coordinates of the ND in the AFM topography image for further indentation experiments (details of the AFM-confocal correlation microscopy are given in Supplementary Fig. 6).

The indentation experiment was carried out by descending the AFM tip into the PDMS film in the proximity of the ND, with a programmed indentation sequence from location #1 to #92 (red dots in Fig. 2a). A constant loading force $P = 150$ nN was applied during the indentation. The ODMR spectra of the ND under two different magnetic fields were measured with and without the indentation on each location from #1 to #92 (typical ODMR spectra plotted in Supplementary Fig. 7). The rotation of the ND was derived from the ODMR spectra for various displacements $\rho$ $(\rho, \theta)$ of the ND from the AFM indentation positions, shown in the figure (Fig. 2b), in which the arrows represent the projected rotation axes and the rotation angle. For all indentation locations, the polar angle $\theta_r$ of the rotation axis was about 90° and the azimuthal angle $\phi_r$ was about $\theta - 90°$ (Fig. 2c), which indicates that the rotation axes of the ND were approximately on the $x$–$y$ plane and perpendicular to the relative displacement $\rho$ between the ND and the AFM indentation position.

The deformation due to the indentation by an AFM tip applied to the origin (0,0) was reconstructed from the rotation data of the ND as a function of the displacement from the AFM tip, as shown in Fig. 2d, using the fact that the PDMS film is homogeneous on the $x$–$y$ plane (see Supplementary Fig. 5b). Figure 2d shows an axisymmetric characteristic about the origin.

We further show the rotation angle $\chi$ (Fig. 3a) and the reconstructed surface $z$ (Fig. 3b) as functions of the distance $\rho$ between the ND and the AFM tip. Similar trends were observed in all measurements on NDs at different locations on PDMS (data of three NDs shown in Fig. 3 and Supplementary Note 4), which is consistent with the homogeneity of the PDMS film. The standard deviation of $\chi$ was $\sigma_\chi = 0.5°$ and in turn the standard deviation of the reconstructed $z$ value was $\sigma_z = 5.2$ nm, which defines a nanometer precision of deformation measurement (details of data fluctuation are given in Supplementary Fig. 10).

The rotation and deformation data are best described with linear elastic bilayer numerical simulations[40]. The model consists of an oxidized surface layer of thickness 500 nm on the top of bulk PDMS. The oxidation layer resulted from the plasma treatment of the PDMS sample[44,45]. The 500 nm thickness of the oxidation layer was estimated from the AFM stiffness mapping of a cross-sectional PDMS sample (see Supplementary Fig. 5a). In the simulation, the apparent Young's modulus of the bulk PDMS was ≈0.78 MPa (as measured by AFM indentation, see Supplementary Note 4), and the Young's modulus of the oxidation layer

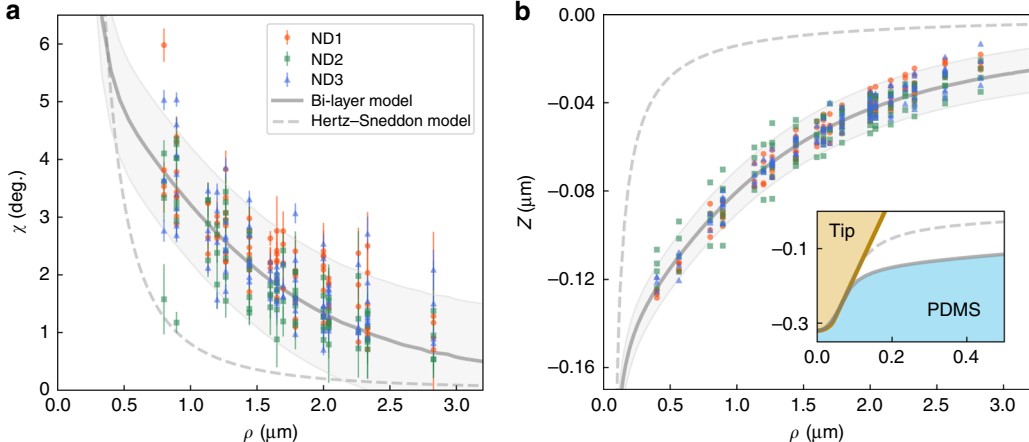

**Fig. 3** Comparison between the nonlocal deformations from experiments and modeling. **a** The rotation angle and **b** the reconstructed deformation as functions of the distance between the nanodiamond (ND) and the atomic force microscopy (AFM) tip. The solid lines are the simulation results of the bilayer model, with the shadowed regions bounded by the standard deviations $\sigma$ as $\chi \pm 2\sigma_\chi$ (1°) and $z \pm 2\sigma_z$ (10 nm) in **a** and **b**, respectively. The dashed lines are simulation results from the Hertz–Sneddon model. The inset in **b** zooms in the simulation results near the AFM tip. The vertical error bars in **a** are the fitting errors

was about 9.4 MPa (about 12 times greater than the bulk value, which is consistent with the literature[45]). The simulated rotation angle $\chi$ of the ND and the deformation $z$, plot respectively in Fig. 3a, b, are well consistent with the experimental data. In addition, the simulated depth-loading curve with the same parameters also matches the experimental result (see Supplementary Fig. 12b). In contrast, the conventional Hertz–Sneddon model, which does not contain the oxidation surface layer, led to simulation results (dash gray lines in Fig. 3) deviating significantly from the experimental data.

Conventionally, the mechanical properties of materials can be studied by measuring the local deformation profile using indentation-based methods (e.g., AFM), but the mechanical parameters of the material can only be determined when a correct model is employed, and the choice of the model relies on accurate knowledge of tip parameters and indentation conditions, the determination of which is not straightforward in many cases. As a result, one would not be able to validate the choice of contact models and consequently determine the mechanical properties of samples based on the local deformation profiles. For example, if one considers Johnson–Kendall–Roberts (JKR) model[46] and Derjaguin–Muller–Toporov (DMT) model[47], which are commonly used for studying adhesive materials, the two models generate similar fitting for a local deformation profile, though with different mechanical parameters (see Supplementary Note 6). On the contrary, the non-local deformation profile can provide additional information for validating the specific contact models (Supplementary Note 6).

**Non-local deformation mapping—multi-ND method.** The second protocol utilized multiple NDs to map the deformation around the indentation imposed at one spot. We demonstrated this multi-ND sensing of deformation induced by a single nanoindentation on a gelatin particle in water (Fig. 4a). By measuring the rotation of the NDs, the deformation of the gelatin particle was reconstructed.

We synthesized gelatin microparticles and dispersed the NDs on the surface in water (for details see Methods). Figures 4b, c show the AFM and fluorescence images of a gelatin microparticle with NDs docked on the surface. The white dashed circle marks the boundary of the gelatin particle, which was about 30 μm in diameter and about 10 μm in height. The comparison between the AFM and the fluorescence images was used to identify the

displacements of the NDs from an AFM indentation (details can be found in Supplementary Note 1). Figure 4d shows the fluorescence image of an examined sample area. The rotation of the NDs is plotted on the same image, with the directions of the arrows representing the rotation axes projected to the $x$–$y$ plane and their lengths representing the rotation angles.

The rotation measurements were carried out on eight groups of NDs (Group I–VIII) around eight different indentation locations. In all cases, the rotation axes of all measured NDs were found approximately on the $x$–$y$ plane and the NDs all rotated toward their respective indentation locations (see Supplementary Fig. 15). Figure 4e, f plot the rotation angles of the NDs in Group I–VIII and the corresponding deformation upon the respective indentations, as functions of the distance between the NDs and the indentation locations (see Methods).

The high precision and the small AFM tip allowed us to study an important effect in deformation upon loading—the effect of surface tension. The surface tension has recently been identified as a key factor in affecting material deformation upon loading[19–23]. The elastocapillary phenomenon has been disclosed as an important reflection of competition between the bulk elastic energy and surface energy[23]. However, the elastocapillary phenomenon only becomes significant when the scale of deformation is smaller than the elastocapillary length scale $L = \tau^0/E$ ($\tau^0$ is the surface tension and $E$ is the Young's modulus). To have an observable elastocapillary effect, one either requires that the materials be very soft (with low $E$) and/or have high surface tension (large $\tau^0$), or is able to measure a small deformation to high precision (for a small indentation caused by a small tip with a weak loading).

In our experiment, the indentation was induced by an AFM tip with a tip radius ~0.15 μm and with a small loading force of 15 nN. The deformation measurement based on ND rotation sensing allowed the small deformation to be measured to a high precision. Thus, we were able to perform a quantitative comparison between the experimental data and the elastocapillary model. We employed an elastic model including the effect of the surface tension to simulate the gelatin deformation upon an AFM indentation[48] (see Methods and Supplementary Note 8). The simulated results are shown in Fig. 4e, f. The simulation assumed the Young's modulus $E = 6.5$ kPa and a surface tension $\tau^0 = 6.5$ mN m$^{-1}$ for the gelatin particle measured in water. These parameters were consistent with the literature on similar material

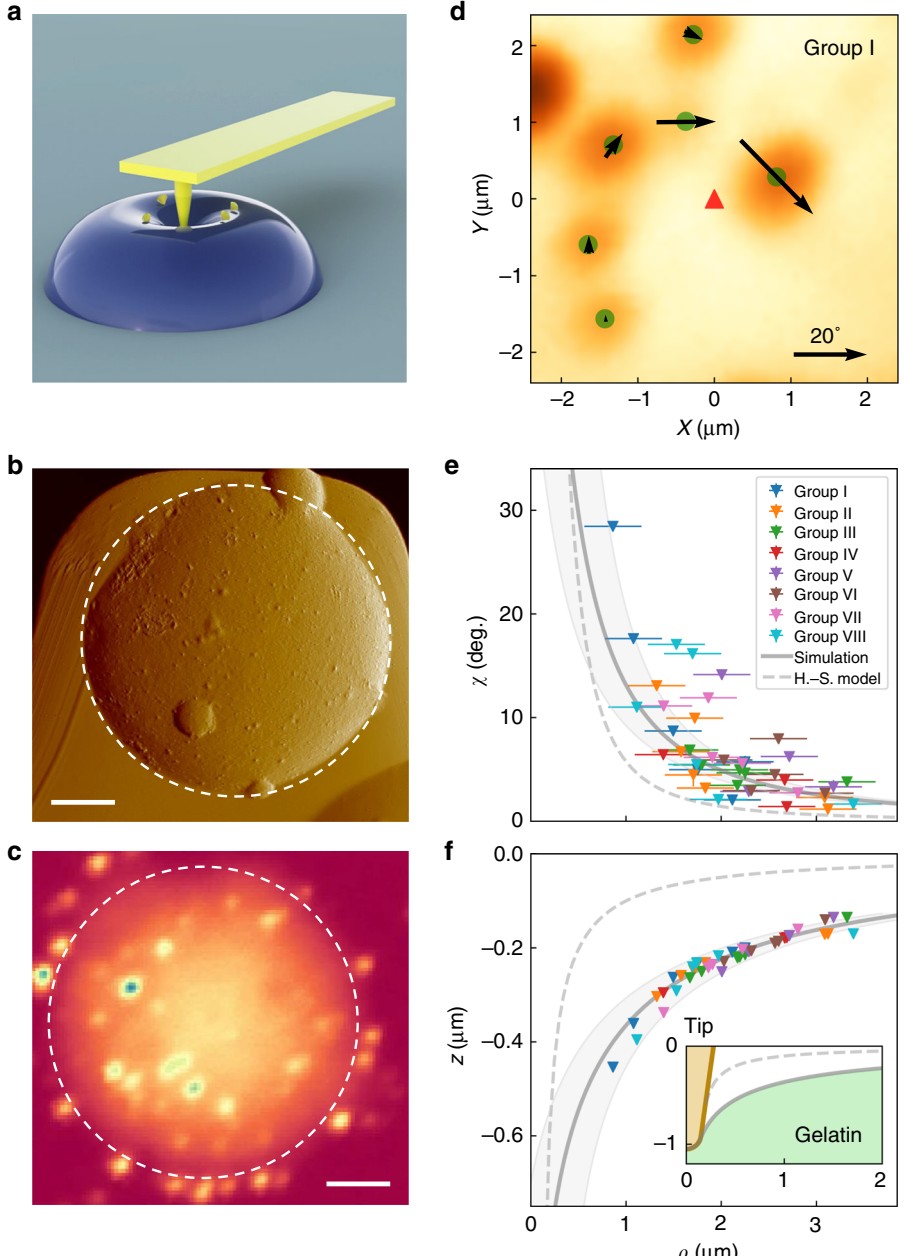

**Fig. 4** Multi-nanodiamond (ND) mapping of deformation induced by a single indentation. **a** Schematic of the gelatin deformation by an atomic force microscopy (AFM) indentation. The NDs were dispersed on the surface of the gelatin. **b** The AFM image, and **c**, the fluorescence confocal image of the gelatin sample. Scale bars in **b** and **c** are 5 µm. **d** Fluorescence confocal image showing the NDs in Group I. The green dots mark the NDs indicated by the highest fluorescence intensity. The directions and the sizes of the arrows represent the rotation axes and the rotation angles of the NDs, respectively. The red triangle marks the position of the AFM tip indentation. **e** The rotation angle $\chi$ and **f** the reconstructed deformation taken from all eight groups of NDs (around eight different AFM indentation positions), plotted as functions of the distance between the NDs and the AFM tip indentation positions. The data from the NDs in the same group are presented in same color. The solid gray lines are the simulation results using the surface tension model (with the gray shadow showing a ±300 nm uncertainty of distance given by the optical resolution). The dashed lines are simulation results from the Hertz–Sneddon model. The inset zooms in the simulation results of the gelatin surface near the AFM tip. The horizontal error bars in **e** are the optical resolution

systems[49,50] and were also confirmed by the agreement between the experimental data and the numerical simulation of the depth-loading profile (Fig. S18b). The surface tension effect was clearly observed in our deformation measurements. Indeed, the simulation without the surface tension effect (Hertz–Sneddon model, gray dashed lines in Fig. 4e, f) deviates significantly from the experimental data. It is worth noting that the models with and without surface tension are differentiated only by the non-local but not the local deformation profile. Both models fit well the

local depth-loading profile (see Fig. S18b), though with different fitting parameters. In other words, validation of the contact model is realized using the non-local deformation profile.

The identification of the surface tension in the gelatin measurement and the model validation are important for studying the mechanical properties of soft matters. In fact, recent studies[19–22] on elastocapillary phenomena suggest that a new model is needed to interpret indentation measurements on cells, where the surface tension plays a significant role but is not

included in the extraction of the mechanical properties of the cells. The correct modeling would be important to evaluating the stiffness of cells for, e.g., differentiating cancerous ones from normal ones[51,52]. As a model system, gelatin was used in our experiments with a Young's modulus comparable to that of the plasma membrane (a few kPa). Our results provide the first set of experimental data obtained in aqueous environment and demonstrate the role of surface tension in determining its mechanical properties, paving the way for future investigation of mechanical properties/mechano-stimuli-induced response of live systems.

## Discussion

In conclusion, we have developed a sensitive method to measure nonlocal nanoscale deformation by combining ND orientation sensing and AFM indentation. The technique features a high precision (~5 nm) in deformation along the loading direction, which is one or two orders of magnitude higher than that of the conventional fluorescence-based methods. The deformed surface profile obtained with high precision contains abundant information of the material mechanical properties, leading to the disclosure of heterostructures in PDMS films (due to surface oxidation), and to the first measurement of the elastocapillary effects at the gelatin/water interface.

The two protocols, namely, the single-ND method and the multi-ND method, have complementary advantages and disadvantages. The single-ND method features a nanometer precision (~5 nm) in the loading direction, as determined by the rotation angle measurement precision, and sub-hundred nanometer lateral resolution, as determined by the AFM tip size or the scanning step size, whichever is larger. However, the deformation reconstruction based on the single-ND method relies on the assumption that the material is laterally homogeneous. For deformation reconstruction of laterally inhomogeneous materials, the multi-ND method is needed. The lateral resolution of the multi-ND method is determined by the optical resolution, or the density of the surface ND, whichever is larger. The sample preparation method in our current experiments results in a random dispersion of the NDs without much control on their density and locations. In the future, the surface functionalization of the NDs to achieve high dispersity, as well as control on the anchoring site, may be employed to improve the deformation reconstruction. Then the ultimate limit of the lateral resolution is the optical resolution.

This work holds great promises in future investigation of soft materials, including live cells. The low cytotoxicity[53,54] and high photostability of NDs and long spin coherence time of NV centers at room temperature[27] make NDs attractive for a variety of bioapplications. Indeed, NDs have already been used for bioimaging/-tracking[30,32,55] and drug delivery[56,57]. For potential biosensing applications, time resolution of 1 s is achievable with the current method (for time resolution estimation, see Supplementary Note 9). Although such timescale (second) is insufficient to study many dynamic behaviors of live systems, it may still be useful for single-site investigation of some physiological processes in live systems.

## Methods

**Setup and sample preparation**. We fabricated an Ω-shape microwave antenna on a glass slide and then spin coated PDMS on the top. The thickness of the PDMS was ≈50 μm. When the PDMS was degassed, we modified the PDMS surface with O₂ plasma. The dilute ND solution (2 μg mL⁻¹ concentration) was thus spin coated on the PDMS film.

Gelatin (1.5 g) was dissolved in 10 mL of deionized water at 60 °C. The 0.15 g mL⁻¹ gelatin solution was added to the Span80 solution and the mixture was vigorously shaken for 20 s. After the shaking, the gelatin in toluene emulsion was stirred at 300 rpm and cooled under ambient conditions for 1 h. Then, the

solidified particles were sonicated for 2 min. Finally, the particles were washed three times with 30 mL of acetone, and then three times with 30 mL of deionized water. The synthesized gelatin microspheres were spin coated onto a confocal dish with a microwave antenna (a 20-μm copper wire was bonded on the surface of a glass slide before a holder was glued on the glass slide to have a liquid chamber with microwave access). Then the ND solution was dispersed on gelatin microspheres using the drop-casting method.

The setup was a homebuilt confocal microscope correlated with an atomic force microscope (see Supplementary Note 1 for details). In the PDMS indentation experiments, two magnetic fields were formed by combinations of two permanent magnets (one was fixed, and the other one was movable between two positions, so that the magnetic field was switchable between **B** and **B'**). In the gelatin indentation experiments, the magnetic field switching was implemented by a permanent magnet and an electric magnet (in which the current alternated its direction and kept its magnitude constant to avoid the heating difference).

**AFM indentation**. We calibrated the AFM setup by performing indentation measurement on sapphire and thermal tune to extract the tip deflection sensitivity and the spring constant, respectively. The tip snap-in point was taken as the contact point and the cantilever deflection was subtracted using pre-calibrated cantilever parameters. The tips we used were made of silicon nitride, whose Young's modulus (in the GPa range) is much larger than those of our samples (in the range of kPa to MPa). As a result, the tip deformation is negligible as compared with the sample deformation. With the contact point set as the snap-in point and the tip deformation taken as zero, we estimated the depth of indentations as the distance between the contact points and the force set points.

In the single-ND method where PDMS was selected as the model sample, a DNP10-A AFM tip (calibrated spring constant: 0.39 N m⁻¹ and deflection sensitivity: 42.28 nm V⁻¹) was used, and the maximum indentation force was controlled at 150 nN. In the multi-ND method where gelatin was employed as the model sample, a PFQNM-LC-A AFM tip (calibrated spring constant: 0.089 N m⁻¹ and deflection sensitivity: 21.01 nm V⁻¹) was used, and the maximum indentation force was controlled at 15 nN.

To avoid possible artifacts induced by the tip–ND interaction, the NDs were chosen not too close to the indentation location. In this study, the closest tip–ND separation was at 800 nm, with the information of deformation near the tip missing. In the future, cylindrical-shaped tips, which have a smaller interaction range with the NDs, may be used to access deformation closer to the tip.

**ND preparation**. ND solution was purchased from Adamas Nanotechnologies. Each ND (with typical diameter of 150 nm) has about 500 NV centers. The ND solution was diluted in aqueous solution to 2 μg mL⁻¹. The dynamic light scattering and transmission electron microscopy measurements showed good dispersion property of NDs (see Supplementary Fig. 6). Before dispersion of NDs on the sample surface, we sonicated the ND solution for 10 min to avoid aggregation of NDs on the surface and finally check the ND distribution by confocal scan and ODMR measurements.

**Tracking ND rotation by ODMR spectra**. The normalized ODMR spectra under the two known magnetic fields are written as[25]

$$S(f) = b - \sum_i C_i \left[ \frac{\Delta f^2}{4(f - f_i^+)^2 + \Delta f^2} + \frac{\Delta f^2}{4(f - f_i^-)^2 + \Delta f^2} \right], \quad (1)$$

where $b$ is the baseline, $C_i$ represents the ODMR contrast of the NV centers along the $i$th crystallographic orientation, $\Delta f$ is the linewidth (FWHM), and $f_i^\pm$ are the transition frequencies. Instead of fitting the transition frequencies, we first fitted the ODMR spectra under the two magnetic fields before the rotation, with the fitting parameters being the initial ND orientation (denoted as the Euler angles ($\alpha$, $\beta$,$\gamma$), see Supplementary Note 2–1 and Fig. S2b), the baselines, the contrasts, the FWHM, and the shift of the zero-field splitting $\Delta D$ (slightly different among different NDs). Then, the rotation of the ND was determined by fitting the two ODMR spectra after the rotation with the rotation axis and angle ($\theta_r$,$\phi_r$,$\chi$), the contrasts, and the baselines as the fitting parameters. The Euler angles, the FWHM, and $\Delta D$ were fixed to be the values obtained by fitting the spectra before the rotation. For each ND, the contrasts and the baselines obtained by fitting the spectra before and after the rotation were approximately the same. For details of the fitting methods, see Supplementary Note 2.2.

**Deformation reconstruction algorithm**. The deformed surface $z(\boldsymbol{\rho})$ was reconstructed by the least-square fitting of the ND rotation data using the gradient field **g** ($\boldsymbol{\rho}$)$=\nabla_\rho z(\boldsymbol{\rho})$[41]. The normal vector of the deformed surface is **n** = $(-g_x,-g_y,1)$, which is related to the rotation of the attached ND as $\hat{\mathbf{n}}(\boldsymbol{\rho}) = \mathbf{R}(\theta_r, \phi_r, \chi)\hat{\mathbf{z}}$, where **R** is the rotation matrix with rotation axis $(\theta_r,\phi_r)$ and rotation angle $\chi$, and $\hat{\mathbf{z}}$ is the $z$-axis. Therefore, the gradient field was deduced from the rotation of the ND by $\mathbf{g}(\boldsymbol{\rho}) = (-\frac{\hat{n}_x}{\hat{n}_z}, -\frac{\hat{n}_y}{\hat{n}_z})$. For the gelatin experiment, with the assumption that the deformation was axisymmetric, the measured gradient field in the polar coordinate

was $\left(g_\rho, g_\theta\right) = (\tan\chi(\rho), 0)$. The deformed surface was reconstructed by integration $z(\rho) = \int_0^\rho \tan\chi(\rho')d\rho'$. For details, see Supplementary Note 4 and 7.

**Numerical simulation of deformation**. The indentation-caused deformation of the PDMS film was simulated with a half-infinite linear elastic model, with a harder coating layer under an axisymmetric loading[40]. The Young's modulus of the PDMS and the oxidation layer were chosen as 0.78 MPa and 9.4 MPa, respectively, the Poisson's ratio was 0.33, and the thickness of the oxidation layer was 500 nm. The chosen parameters are consistent with the literature[45,58] and with our AFM measurement. For details see Supplementary Note 5.

In the gelatin case, we used a half-infinite axisymmetric linear elastic model, including the effect of the surface tension[48]. The Young's modulus was chosen as 6.5 kPa, the Poisson's ratio was 0.5, and the surface tension of the water–gelatin interface was 6.5 mN m$^{-1}$. The chosen parameters agree with the literature[49,50]. For details see Supplementary Note 8.

## Data availability
The data which support the findings of this work are available upon request from the corresponding authors.

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

## Acknowledgements

We would like to thank the discussion with Ting Zhang for the gelatin synthesis; and Sen Yang for the setup construction. This work was supported by Hong Kong RGC/CRF (C4006-17G), and CUHK Group Research Scheme under project code 3110126.

## Author contributions

Q.L. and R.B.L. conceived the idea and supervised the project. K.X., C.F.L., W.H.L., R.B.L. and Q.L. designed the experiments. K.X. and C.F.L. constructed the setup. C.F.L. and K.X. performed the experiments. W.H.L., C.F.L., K.X., R.B.L. and Q.L. analyzed the data. W.H.L. designed the rotation tracking protocol and the reconstruction algorithm, and performed the numerical simulation. M.H.K. synthesized the gelatin. Z.Y.Y. and X.F. synthesized the PDMS film. K.X., C.F. L., W.H.L., R.B.L. and Q.L. wrote the paper and all authors commented on the paper.

## Additional information

**Competing interests:** The authors declare no competing interests.

