## [Peer Review File · Nature Communications]

Reviewers' comments:

Reviewer #1 (Remarks to the Author):

The manuscript describes a smart way to reconstruct the deformation of the surface of a soft sample produced by the application of a nanoscale load using the tip of an AFM. The "trick" is the analysis of the rotation of a nanodiamond (ND) particle (or many ND particles) which is docked on the surface by drop casting. The authors discuss two methods: in the first a single ND is used and the AFM tip probes points around it; in the second method, AFM indentation is performed in a single point and several ND are analyzed. Although it may be hardly interesting for a broad readership, I think that the work is interesting for experts in the field and its presentation is definitely eye-catching, even if not to clear in some points. I think that manuscript could be suitable for publication, but I would recommend the authors to address some (minor) issues to improve the significance of their work:

- As a general comment, the authors should check the English and the coherency of information given in the text. For instance, the maximum load is reported as 150 nN in the discussion while it is 15 nN the conclusions. Also, in the conclusions the tip radius is reported as 100 nm. This information is not presented before and is somehow surprising as generally AFM tip radii are much smaller, say one order of magnitude smaller. Is the result of purposely worn tip? Have they determined it using, e.g., electron microscopy? Or did they consider only the shape of the reconstructed indentation?
- The authors demonstrated their methods on two specific cases with a not so evident scientific interest. This is correct of course as they report a proof of concept. Nevertheless, I think the authors should stress the possibility given by their methods. Indeed, the method using a single ND allow only the determination of the indentation profile in case of homogeneous materials. Could this be interesting to validate specific contact models? The authors consider Hertz and layered materials, but they could expand the discussion (e.g., suitability of JKR versus DMT contact models, which is still a discussed topic in AFM indentation...).
- The second method is suitable for not-homogeneous materials. Here, I see a limitation in the fact that the investigated locations are randomly selected (those in correspondence of which a ND is docked on the surface). A honest discussion of the limits of the methods should be implemented.
- Even if in a synthetic way, the authors could discuss some practical issues, like the effect of surface morphology and the possibility (if any) that the presence of the cantilever can shield the area and thus can limit the capability to investigate the sample close to the indentation point.
- As for the lateral resolution, the authors could discuss the limit of resolution they can predict. I imagine that lateral resolution depends on both the AFM parameters (especially in the first method) and on the ND particle dimensions.

Reviewer #2 (Remarks to the Author):

Overall comments

This manuscript made an important attempt to characterise deformations and their distribution in the context of biological systems (and beyond). Mechanobiology is now spearheading discoveries in biomedical sciences, thus any work that contributes to better understanding of mechanobiological responses that regulate the function of biological structures, is critically important. Therefore, this manuscript is definitely of high importance – high impact (I prefer to use term high-influence, impact is just a point, typically short term changes, influence changes the course of science in long term).

Abstract need to be somewhat better, at the moment significance of this work and WHAT IF is not really presented and needs to be rewritten to show the strength of presented results and their application.

Introduction; I would be very careful with the statement such as "...localized deformation at the contact region can be routinely obtained..." it is not correct because the deformation here is a sum

of the deformation of the sample and deflection of the cantilever and very much depended on the conditions of the indentation (speed, liquid type, humidity ...) and type of indenter. Also the fact that biological sample is typically viscoelastic makes such measurements 'condition-dependent'. Please check a few papers with mathematical and experimental modelling of indentation with AFM (Alexander L. Svistkov, Igor Sokolov...).

I fully agree with authors that there are several methods (well-established) to look at the local deformation, the good example is tracking cells movements and stresses using nanoparticles (<http://dev.biologists.org/content/143/2/186>) but the resolution could be a problem. Therefore proposed methodology advances substantially our capability to look at the deformation in materials, including biological systems.

Presented concept of using diamond particles with vacancies is very interesting. While nanodiamond raises several controversies in the context of its biological/medical applications, if we use it on 'fixed' system there is no problem with ND; its application to living systems may not be recommended due to its potential toxic effects of cells and proteins. Please make it clear in your manuscript, how you recommend doing measurements, for which systems and why. Also please comment how you avoided aggregation of nanodiamond? When aggregated (which is unavoidable and in fact extremely difficult to have ND as monodisperse solution without aggregates) sizes are not well controlled, what was the size of your aggregates? And how robustly it was attached to the surface?

Concluding parts of intro must be much better and showing how we can actually use this method (how easy/difficult, time consuming it is?) and why we need this. I do not agree with 'potentially dynamical behaviours of systems under mechanical stimuli.' how do you want to achieve this – please comment and give examples.

Result section is exceptionally well presented and written. However, authors use wrong nomenclature in some places; e.g. AFM indentation does not allow to measure Young's modulus, it is Apparent Young's modulus! Another question would be how did you make sure that the tip did not penetrate to the gel/polymers? There is a likelihood that it could have happened and perhaps selection of colloidal probe could be recommended, did you consider that? You did not show indentation curves to be able to see that so I cannot comment – could you please add them to supplementary file?

The other aspect I wanted to ask you to comment on is that both PDMS and Gelatin are homogenous only in macroscale but not on micro and definitely not on nanoscale, how do you account for the fact that chains of both polymers are oriented and entangled in random directions and this may impact on the local deformation – also potentially movements of NDs during deformation. Can you in fact detect this using your method? This would be what we need in testing biological systems. Please comment.

Another aspect which is rather minor, why did you use Hertz model? It is not right model in AFM indentation with sharp tip!

Methods; I cannot see the method for AFM indentation, we need to see which tip you used (spring constant) this is critical part to say whether you actually could indent to depth that you claim. As previously mentioned AFM does not allow to measure depth of the indentation but it allows to estimate it if we have well-defined experimental setup and modelling protocol in place. Please comment on that.

Overall, exceptionally good science with outstanding results, which were presented well. Small parts need better clarification; which ND type/size was used, how did you ensure individual ND are attached to the samples, if aggregates how this could influence your results since they are oriented in direct directions (is this a problem – please clarify and comment). AFM and ND preparation sections are missing.

This manuscript needs much better justification why we need this results/science, and what is the significance of this work. How we can benefit from it? Who need that?

Minor comment; several editorial glitches were detected but I know that of accepted by the journal it will be all fixed by professional team.

Reviewer #3 (Remarks to the Author):

"Nanometer-precision non-local deformation reconstruction using nanodiamond sensing", by Xia et al.

The authors demonstrate measurements of surface deformation with the use of the NV colour centre in diamond. The advantage of this technique is shown to rest in the ability to measure these surface deformations at positions away from a probing indenter allowing the mechanical response to be measured in more detail than previously possible. This compliments techniques like nano-indentation where the sensor is the probe.

The concept of the technique is relatively straight-forward to understand and is a unique extension of the utility of the NV centre in diamond. I am of the opinion that this paper warrants publication in Nature Communications. However, the clarity could be greatly improved first as the style and minor grammatical errors detract from work. The sections of the manuscript where the meaning was not clear are discussed below.

Comments:

General:

1. The manuscript describes the new technique quite well using PDMS and gelatin particles as model systems. Of course the focus of the paper is on the implementation of the technique. However, the paper may have broader appeal if a short discussion on the uniqueness of the data relative to other techniques and the new insights it provides is included.

For example, the authors state that the technique reveals the significance of surface/interface effects in material deformation (line 30). Indeed, it is shown that surface tension is required to understand the results. However, the significance and implications of these results are not put into context making it difficult for readers without a the relevant background to appreciate.

2. Although it is a technical consideration it may be beneficial to include an indication of how long it takes to make this measurement. This might be important for some more dynamic systems as mentioned on line 86 ("dynamical behavior of systems under mechanical stimuli.").

3. The Hertzian model appears to describe the indenter force-depth traces in Fig. S12 and Fig. S17 well. Please clarify why the same model is not adequate to describe the data in Fig. 3 and 4 of the main text.

Typographical errors:

Line 20:

Acronyms are not defined in the abstract.

Line 28:

"This approach features a 5 nm precision in the loading direction and a sub-hundred nanometer lateral resolution."

Does this simply mean 5 nm precision supplied by the AFM normal to the surface whereas the sub-hundred nano-meter resolution is the diffraction limit and/or the size of the nano-diamond determined by the AFM scan? With the multi-ND approach the resolution is also determined by the ND density which should also be greater than the diffraction limit to avoid convoluted spectra. Could this be concisely clarified further in the text.

Line 51:

"Deformation of such large scales may miss the deformation characteristics at finer scale,..."

Suggested rewrite:

"Such techniques lack the sensitivity required to probe important deformation phenomena such as the effect of surface tension at the interface between soft materials and liquids.

Line 64:

"When a nanodiamond is docked on a material surface,..."

I recommend replacing the term "dock" with an alternative. The various common usages of this word may cause confusion. For example, this sentence may read:
Changes in the rotation of a ND placed on a material surface is directly related to changes in the surface gradient and hence its deformation.
They may then be referred to as "surface NDs" to be succinct.

Line 70:

Use "ND" or "nanodiamond" consistently.

Line 71:

.."multiple NDs docked at various positions..."
"multiple NDs located at various positions".

Line 85:

"which features a precision of < 10 nm,..."

What does this value refer to? deformations provided by the indenter? Or the accuracy in determining the surface gradient (as described on line 179)?

Line 92:

"NDs are docked on the material surface."

NDs are distributed across the material surface.

Line 183:

"The rotation and deformation data are well consistent with the numerical simulation based on a linear elastic bi-layer model."

Consider rewriting this:

"The rotation and deformation data are best described with linear elastic bi-layer numerical simulations."

List of Changes:

All changes are marked in shading in the manuscript and supplementary material for easy reference.

1. General

- (a) In the author list, the second affiliation “*Centre for Quantum Coherence*” is replaced by “*The Hong Kong Institute of Quantum Information Science and Technology*”.
- (b) We have modified the English in the revised manuscript and revised supplementary information.
- (c) Labels of the scale bars in the figures are removed and the corresponding descriptions are added in the captions.
- (d) Descriptions of the error bars in the figures are added in the captions.

2. Abstract section:

- (a) On page 2, lines 24-25, we add “*Moreover, interpretation of the local measurement relies on correct modeling, the validation of which is not straightforward..*”
- (b) On page 2, line 27, we replace “*ND*” with “*Nanodiamond*”.
- (c) On page 2, line 27, we replace “*AFM*” with “*Atomic force microscopy*”.
- (d) On page 2, line 29, we delete “*Using this tool, we mapped out the deformation of a PDMS thin film in air and that of the gelatin microgel particles in water, with*” for the limitation on the number of words in the abstract.
- (e) On page 2, lines 29-31, we add “*The non-local deformation profile can validate the models needed for mechanical property determination.*”.
- (f) On page 2, lines 32-33, we add “*The non-local nanometer-precision sensing of deformation facilitates studying mechanical response of complex material systems ranging from impact transfer in nanocomposites to mechano-response of live systems*”.
- (g) The abstract is revised for the limitation on the number of words.

3. Introduction section:

- (a) On page 3, lines 40-47, we replace “*With the development of pico-indenters and the advances of atomic force microscopy (AFM) based methods, localized deformation at the*

contact region can be routinely obtained and measured with lateral spatial resolution down to tens of nanometers¹¹⁻¹³ with “Although local deformation at the contact region can be obtained by well-established methods¹¹⁻¹⁴, the resolution of the measurement remains unsatisfactory. More importantly, the measured local deformation results from a summation of sample deformation and cantilever deflection. It largely depends on the conditions of the indentation such as the type of indenter, the indentation speed, the type of liquid employed, the humidity, etc^{15,16}. If soft matter or biological systems are studied, the viscosity of the samples also plays a role. Using only local deformation profiles is usually inadequate to determine the material mechanical properties.”.

- (b) On page 3, lines 55-57, we replace *“Deformation of such large scales may miss the deformation characteristics at finer scale, which are essential to understanding many important deformation phenomena (such as the effects of surface tension on deformation at the interfaces between soft materials and liquids).”* with *“Such techniques lack the sensitivity required to probe important deformation phenomena such as the effects of surface tension at the interfaces between soft materials and liquids.”.*
- (c) On page 4, lines 59-60, we add *“Such a tool would not only enable the investigation of impact transfer at nanoscale in heterostructured materials with multiple phases and interfaces,”* .
- (d) On page 4, lines 69-70, we replace *“When a nanodiamond is docked on a material surface, the deformation is related to the orientation of the surface and hence the rotation of the nanodiamond.”* with *“The deformation is related to the orientation of the surface and hence the rotation of the nanodiamonds (NDs) that are situated on the surface.”* .
- (e) On page 4, lines 72-73, we replace *“nanodiamonds docked on a surface”* with *“NDs on a surface”*.
- (f) On page 4, line 78, we replace *“docked”* with *“located”*.
- (g) On page 5, lines 81-82, we replace *“an in-plane spatial resolution limited by the ND size (which is ~200 nm in our experiments).”* with *“an in-plane spatial resolution ~200 nm (as determined by the AFM parameters - the scanning step size in the present experiments).”*.
- (h) On page 5, line 85, we delete *“The surface tension at soft materials/water interface is expected to play an important role in determining the deformation characteristics when the perturbation is weak³⁷.”* .

- (i) On page 5, lines 86-89, we add *“This is particularly relevant to the mechanical property measurement of soft matters such as single cells, when accurate determination of the stiffness cannot be done without knowledge of the elastocapillary effect^{19-22,40}.”*
- (j) On page 5, lines 92-93, we replace *“potentially dynamical behaviors of systems under mechanical stimuli.”* with *“paving the way for investigating mechanical properties/mechano-stimuli induced response of live systems in the future. ”*

4. Results section:

- (a) On page 5, line 98, we replace *“docked on”* with *“distributed across”*.
- (b) On page 8, line 140; page 12, line 199; page 16, line 256; page 12, caption and legend of Fig.3; and page 14, caption and legend of Fig.4, we replace *“Hertz model”* with *“Hertz-Sneddon model”*.
- (c) On page 9, line 156, we move *“NDs with a typical size of ~200 nm were dispersed on the surface of PDMS film by spin-coating.”* to line 158-159 and replace *“~200 nm”* with *“~150 nm”*.
- (d) On page 9, lines 160-162, we move *“The homogeneity of the PDMS film was confirmed by the almost identical load-depth profiles obtained at different locations on the sample by AFM indentation (See SI Fig. S5b).”* from line 185.
- (e) On page 11, lines 189-190, we replace *“The rotation and deformation data are well consistent with the numerical simulation based on a linear elastic bi-layer model³⁶”* with *“The rotation and deformation data are best described with linear elastic bi-layer numerical simulations³⁹”*.
- (f) On page 11, line 193, we replace *“Young’s modulus”* to *“apparent Young’s modulus”*.
- (g) On page 12, lines 202-213, *“Conventionally the mechanical properties of materials can be studied by measuring the local deformation profile using indentation based methods (e.g. AFM), but the mechanical parameters of the material can only be determined when a correct model is employed, and the choice of model relies on accurate knowledge of tip parameters and indentation conditions, the determination of which is not straightforward in many cases. As a result, one would not be able to validate the choice of contact models and consequently determine the mechanical properties of samples based on the local deformation profiles. For example, if one considers Johnson-Kendall-Roberts (JKR)*

model⁴⁶ and Derjaguin-Muller-Toporov (DMT) model⁴⁷, which are commonly used for studying adhesive materials, the two models generate similar fitting for a local deformation profile, though with different mechanical parameters (See SI Note 6). On the contrary, the non-local deformation profile can provide additional information for validating the specific contact models (SI Note 6)." is added to discuss the validation of the contact model by using the non-local deformation information.

- (h) One page 15, line 234, we move *"The data from the NDs in the same group are presented in the same color."* to the caption of Figure 4.
- (i) On page 16, lines 257-261, *"It is worth noting that the models with and without surface tension are differentiated only by the non-local but not the local deformation profile. Both models fit well the local depth-loading profile (see Fig. S18b), though with different fitting parameters. In other words, validation of the contact model is realized using the non-local deformation profile."* is added to describe the importance of obtaining the non-local deformation information.
- (j) On page 16, lines 262-272, *"The identification of the surface tension in the gelatin measurement and the model validation are important for studying mechanical properties of soft matters. In fact, recent studies¹⁹⁻²² on elastocapillary phenomena suggest that a new model is needed to interpret indentation measurements on cells, where the surface tension plays a significant role but is not included in the extraction of the mechanical properties of the cells. The correct modeling would be important to evaluating the stiffness of cells for, e.g., differentiating cancerous ones from normal ones^{51,52}. As a model system, gelatin was used in our experiments with a Young's modulus comparable to that of the plasma membrane (a few kPa). Our results provide the first set of experimental data obtained in aqueous environment and demonstrate the role of surface tension in determining its mechanical properties, paving the way for future investigation of mechanical properties/mechano-stimuli induced response of live systems."* is added to discuss the significance of obtain the surface/interface effects in material deformation.

5. Discussion section:

- (a) On page 17, lines 284-287 we replace *"The single-ND approach provides an unprecedentedly high precision (~5 nm) in the direction of loading (z) with a high*

laterally spatial resolution (~200 nm).” with “The single-ND method features an unprecedentedly high precision (~5 nm) in the loading direction, as determined by the rotation angle measurement precision,; and sub-hundred nanometer lateral resolution, as determined by the AFM tip size, or the scanning step size, whichever is larger.”.

- (b) On Page 17, lines 287-290, we replace “*However, it relies on the assumption that the material is laterally homogeneous. For applications on laterally inhomogeneous materials, the multi-ND method is needed.*” with “*However, the deformation reconstruction based on the single-ND method relies on the assumption that the material is laterally homogeneous. For deformation reconstruction of laterally inhomogeneous materials, the multi-ND method is needed.*”.
- (c) On page 17, lines 290-295, we replace “*The multi-ND method suffers from limited in-plane spatial resolution (depending on how dense NDs can be dispersed on the surface for ODMR measurement).*” with “*The lateral resolution of the multi-ND method is determined by the optical resolution, or the density of the surface ND, whichever is larger. The sample preparation method in our current experiments results in a random dispersion of the NDs without much control on their density and locations. In the future, the surface functionalization of the NDs to achieve high dispersity, as well as control on the anchoring site may be employed to improve the deformation reconstruction. Then the ultimate limit of the lateral resolution is the optical resolution.*”.
- (d) On page 18, lines 296-303, we replace “*The methods developed in this paper are expected to offer new approaches to investigating surface effect of the materials as well as live systems in liquid environments upon external weak perturbation.*” with “*The present work holds great promise in future investigation of soft materials including live cells. The low cytotoxicity^{53,54}, high photo-stability, and long spin coherence time at room temperature²⁶ make ND very promising for a variety of bio-applications. Indeed, NDs have already been used for bio-imaging/-tracking^{29,31,55} and drug delivery^{56,57}. For potential bio-sensing applications, time resolution of 1s is achievable with the current method. Although such time scale (second) is insufficient to study many dynamic behaviors of live systems, it may still be used for single site investigation of physiological processes in live systems (for time resolution estimation, see SI Note 9). Although such time scale (second) is insufficient*

to study many dynamic behaviors of live systems, it may still be useful for single site investigation of some physiological processes in live systems.”.

6. Methods section:

- (a) On page 19, lines 328-348, we add the details of AFM indentation in “***AFM indentation***” subsection.
- (b) On page 20, lines 350-356, we add the details of nanodiamonds preparation in “***NDs preparation***” subsection.

7. References:

- (a) New references [14-16], [46-47], and [51-57] are cited.
- (b) The order of the references is updated.

8. Supplementary material:

- (a) On page 8, line 153; page 12, line 215; page 21, caption and legend of Fig. S12; page 22, line 322, 340; page 29 line 427; legend of Fig. S18, we replace “*Hertz model*” with “*Hertz-Sneddon model*”.
- (b) On page 12, line 214; page 22, line 323, we replace “*Young’s modulus*” with “*apparent Young’s modulus*”.
- (c) On page 13, line 217, Note 4, we add “(with *Young’s modulus* ~ 1 GPa)”.
- (d) On page 13, lines 227-231, we add “*Dynamic light scattering (DLS) measurement of the ND solution suggested an average hydrodynamic diameter of 150 nm (Fig. S6a). We also checked the sample using transmission electron microscopy (TEM) (Fig. S6b), when reasonably dispersed NDs were found. The diameters of ND were in the range of a few tens of nanometers to ~ 200 nm, consistent with the DLS measurement.*”.
- (e) On page 14, Fig.S6, we add (a) and (b), corresponding to “*a, Size distribution of NDs measured by dynamical light scattering (DLS).* and “*b, Typical TEM image of NDs.*” respectively.
- (f) On page 14, lines 234-236, we add “*A DNPI0-A AFM tip was applied for AFM imaging and nanoindentation experiments with calibrated spring constant of 0.39 N m^{-1} and deflection sensitivity of 42.28 nm V^{-1} .*”.

- (g) On page 14, lines 238-239, we add “*with the indent and withdraw speed was kept for as $0.3 \mu\text{m s}^{-1}$ for total ramp size of $1 \mu\text{m}$.*”.
- (h) On page 22, lines 326-353, we add “*Note 6. Validation of specific contact models*”.
- (i) On page 23, we add Fig. S13 “*Simulations for JKR and DMT models.*”.
- (j) On page 24, lines 367-368, we add “*silicon nitride cantilever (PFQNM-LC-A, Bruker)*”.
- (k) On page 24, Fig. S14, we add “*a. SEM images of PFQNM-LC-A AFM cantilever used in the gelatin experiments.*”.
- (l) On page 25, lines 368-369, we add “*deflection sensitivity of 21.01 nm V^{-1}* ”.
- (m) On page 25, lines 371-373, we add “*using a PFQNM-LC-A cantilever in aqueous solution with holding force setpoint of 15 nN and indent and withdraw speed of $1 \mu\text{m s}^{-1}$ in total ramp size of $3 \mu\text{m}$.*”.
- (n) On page 30, lines 430-442, we add “*Note 9. Sensitivity of the rotation measurement of the surface ND*”.
- (o) On page 30, we add Fig. S19 “*The standard deviation of the rotation angle σ_χ as a function of the data acquisition time t_m . The line is the linear fitting with zero intercept.*”.
- (p) On page 31, Supplementary References, we add reference [12-14].
- (q) The order of the notes, the references and the figures are updated.

Reviewers' comments:

Reviewer #1 (Remarks to the Author):

1. The manuscript describes a smart way to reconstruct the deformation of the surface of a soft sample produced by the application of a nanoscale load using the tip of an AFM. The “trick” is the analysis of the rotation of a nanodiamond (ND) particle (or many ND particles) which is docked on the surface by drop casting. The authors discuss two methods: in the first a single ND is used and the AFM tip probes points around it; in the second method, AFM indentation is performed in a single point and several ND are analyzed. Although it may be hardly interesting for a broad readership, I think that the work is interesting for experts in the field and its presentation is definitely eye-catching, even if not too clear in some points. I think that manuscript could be suitable for publication, but I would recommend the authors to address some (minor) issues to improve the significance of their work:

- As a general comment, the authors should check the English and the coherency of information given in the text.

Response:

We have modified the English in the revised manuscript and revised supplementary information.

2. For instance, the maximum load is reported as 150 nN in the discussion while it is 15 nN in the conclusions.

Response:

Two different AFM cantilevers were used, and two different indentation forces were respectively applied in the experiments. In the “single-ND” method when PDMS is selected as a model sample, DNP10-A AFM tip (calibrated spring constant: 0.39 N m^{-1} , deflection sensitivity: 42.28 nm V^{-1}) was used, and the maximum indentation force is controlled at 150 nN. In the “multi-ND” method when gelatin was employed as a model sample, PFQNM-LC-A AFM tip (calibrated spring constant: 0.089 N m^{-1} , deflection sensitivity: 21.01 nm V^{-1}) was used, and the maximum indentation force was controlled 15 nN. These have been clarified in the Methods

section of the revised manuscript, see page 20, lines 338-343. We have also modified the details in the revised supplementary information Note 4 and Note 7. See SI Page 14, lines 234-236 and SI Page 24, lines 367-369.

3. Also, in the conclusions the tip radius is reported as 100 nm. This information is not presented before and is somehow surprising as generally AFM tip radii are much smaller, say one order of magnitude smaller. Is the result of purposely worn tip? Have they determined it using, e.g., electron microscopy? Or did they consider only the shape of the reconstructed indentation?

Response:

In the PDMS experiment, the AFM tip radius was ~ 50 nm (DNP10-A, Bruker). In the gelatin experiment, we intentionally chose an AFM tip with larger tip radius (~ 150 nm) and smaller spring constant to cater the soft nature of gelatin. The ~ 150 nm AFM tip radius was determined by the scanning electron microscopy (Figure 1 below). The SEM image of the AFM tip has been added to the revised supplementary information Page 24, Fig. S14.

Figure 1. SEM images of the PFQNM-LC-A AFM cantilever used in the gelatin experiments. The scale bar in the inset of (b) is 1 μ m.

4. The authors demonstrated their methods on two specific cases with a not so evident scientific interest. This is correct of course as they report a proof of concept. Nevertheless, I think the authors should stress the possibility given by their methods.

Response:

The potential applications of the work go to two different directions. One possibility is mapping out impact transfer in non-uniform media at nanometer scale. This would be particularly important in understanding the unique mechanical properties of various nanocomposite materials when multiple phases and interfaces exist. The other possibility is studying mechano-response of live systems such as single cells. The present method would add to the existing biological toolbox a function that can measure the non-local response of live systems upon a local stimulation. In any evolving systems, pushing the time resolution in the future would be important to access their dynamics. We have added the related elaboration in the revised manuscript. See Page 2, lines 29-33; Page 5, lines 92-93; and Page 18, lines 296-303.

5. Indeed, the method using a single ND allow only the determination of the indentation profile in case of homogeneous materials. Could this be interesting to validate specific contact models? The authors consider Hertz and layered materials, but they could expand the discussion (e.g., suitability of JKR versus DMT contact models, which is still a discussed topic in AFM indentation...).

Response:

We have run numerical simulations of the deformation upon nanoindentation using both JKR and DMT models [D. Maugis, J. Colloid Interface Sci. 1992, 150, 243], which are commonly considered when studying adhesive materials. The simulation based on the JKR and DMT models gives similar depth-loading curves (Figure 2a), but generates different Young's moduli and adhesion energies ($E_{DMT}/E_{JKR} = 1.02$ and $W_{DMT}/W_{JKR} = 0.6$). Only using the local deformations profiles (e.g. by AFM indentation) one would NOT be able to validate the choice of contact models and consequently determine the mechanical properties of the samples. As a comparison, when non-local deformations are considered, a significant difference is observed

between JKR and DMT models (Figure 2b). The difference between the rotation angles of the surface ND (800 nm away from the indentation point) in the two models is $\sim 3.3^\circ$ (see Figure 2c), which is sufficiently large to be measured by the ODMR. Therefore, the indentation profile determination with high precision by the “single-ND” method provides additional information that can be used to validate the specific contact models. We have added the discussion of the relevant example in the revised manuscript and the revised supplementary information. See Page 12, lines 202-213 in the revised manuscript and Page 22, Note 6 in the revised supplementary information.

Figure 2. (a) Simulation results of the depth-loading curves based on JKR (blue) and DMT (orange) models. The Young’s modulus and the adhesion energy are chosen as $E_{\text{JKR}} = 80 \text{ kPa}$ and $W_{\text{JKR}} = 80 \text{ mN m}^{-1}$ for JKR model, and $E_{\text{DMT}} = 82 \text{ kPa}$ and $W_{\text{DMT}} = 48 \text{ mN m}^{-1}$ for DMT model. (b) and (c) Simulation results of the non-local deformations and the corresponding rotation angles as functions of the distance from the tip based on JKR (blue) and DMT (orange) models. The green line defines the shape of the tip. The indentation force is set as 3 nN (red dot in (a)).

6. The second method is suitable for not-homogeneous materials. Here, I see a limitation in the fact that the investigated locations are randomly selected (those in correspondence of which a ND is docked on the surface). A honest discussion of the limits of the methods should be implemented.

Response:

The second method requires NDs to be pre-dispersed on the sample surface before the indentation measurement. The current dispersion method results in a random dispersion of the NDs without much control on their density and individual locations, limiting the lateral spatial resolution of the deformation measurement. In the future, the availability of small sized ND and their surface functionalization to achieve high dispersity, as well as control on the anchoring site would be important to improve the later spatial resolution of the technique. This has been elaborated in the revised manuscript. See Page 17, lines 290-295.

7. Even if in a synthetic way, the authors could discuss some practical issues, like the effect of surface morphology and the possibility (if any) that the presence of the cantilever can shield the area and thus can limit the capability to investigate the sample close to the indentation point.

Response:

This is indeed a practical limit of the present technique. To avoid any possible artifacts that can be induced by tip-ND interaction, the NDs cannot be placed too close to the indentation location. In the present study, the closest tip-ND separation is maintained at 800 nm, so that information of deformation near the tip is missing. In the future, this can be improved by using cylindrical shaped tip. This has been explained in the Methods section of the revised manuscript. See Page 20, lines 344-348.

8. As for the lateral resolution, the authors could discuss the limit of resolution they can predict. I imagine that lateral resolution depends on both the AFM parameters (especially in the first method) and on the ND particle dimensions.

Response:

The lateral resolution of the “single-ND” method is mainly determined by the AFM parameters, that is, the AFM tip size and scanning step size. As a comparison, the “multi-ND” method suffers from worse lateral resolution, resulting from the fact that the ODMR signals of two NDs can no longer be separated when they are placed within the optical resolution. The lateral

resolution is therefore ultimately determined by the optical resolution. We modified the statement of the resolution in the revised manuscript. See Page 5, lines 81-82; and Page 17, lines 284-287 and lines 290-295.

Reviewer #2 (Remarks to the Author):

Overall comments

1. This manuscript made an important attempt to characterise deformations and their distribution in the context of biological systems (and beyond). Mechanobiology is now spearheading discoveries in biomedical sciences, thus any work that contributes to better understanding of mechanobiological responses that regulate the function of biological structures, is critically important. Therefore, this manuscript is definitely of high importance – high impact (I prefer to use term high-influence, impact is just a point, typically short term changes, influence changes the course of science in long term).

Abstract need to be somewhat better, at the moment significance of this work and WHAT IF is not really presented and needs to be rewritten to show the strength of presented results and their application.

Response:

Abstract has been revised to stress the significance of this work, and also to outlook the “what-ifs”.

2. Introduction; I would be very careful with the statement such as “...localized deformation at the contact region can be routinely obtained...” it is not correct because the deformation here is a sum of the deformation of the sample and deflection of the cantilever and very much depended on the conditions of the indentation (speed, liquid type, humidity ...) and type of indenter. Also the fact that biological sample is typically viscoelastic makes such measurements ‘condition-

dependent'. Please check a few papers with mathematical and experimental modelling of indentation with AFM (Alexander L. Svistkov, Igor Sokolov...).

I fully agree with authors that there are several methods (well-established) to look at the local deformation, the good example is tracking cells movements and stresses using nanoparticles (<http://dev.biologists.org/content/143/2/186>) but the resolution could be a problem. Therefore proposed methodology advances substantially our capability to look at the deformation in materials, including biological systems.

Response:

We have revised the statement in the introduction section. See Page 3, lines 40-46, and Page 5, Lines 92-93.

3. Presented concept of using diamond particles with vacancies is very interesting. While nanodiamond raises several controversies in the context of its biological/medical applications, if we use it on 'fixed'; system there is no problem with ND; its application to living systems may not be recommended due to its potential toxic effects of cells and proteins. Please make it clear in your manuscript, how you recommend doing measurements, for which systems and why.

Response:

The toxicity of ND has been widely studied in the literature. At cellular level, NDs are generally accepted as "non-toxic" based on results obtained from various physiology investigation of cells [S. Yu *et.al.*, *J. Am. Chem. Soc.* 2005, 127, 17604; K. Liu, *et.al.*, *Biomaterials*, 2009, 30, 4249] after being fed with NDs, including viability assays, genotoxic analysis, and immunological responses etc. [V. Vijayanthimala, *et.al.*, *Biomaterials*, 2012, 33, 7794; A. Schrand, *et.al.*, *J. Phys. Chem. B*, 2007, 111, 2]. In fact, NDs have already been widely employed as an imaging agent for bio-tracking [L. McGuinness, *et.al.*, *Nat. Nanotech.* 2011, 6, 358; Simon Haziza, *et.al.*, *Nat. Nanotech.* 2017, 12, 322] as well as a therapeutic agent [Dean Ho, *ACS Nano* 2009, 3, 3825; X. Zhang, *et.al.*, *Adv. Mater.* 2001, 23, 4770; E. Chow, *et.al.*, *Science Translational Medicine*, 2011, 3, 73ra21]. Of course, the toxicity of NDs largely depends on their synthetic origins, surface chemistry, and size. It should also be noted that the long-term systemic toxicity (or non-

toxicity) of NDs remains controversial, since at the system level, safe excretion of NDs largely depends on their size and a number of other factors. In the future, the investigation may start with “fixed” cells, but it is also possible to extend the study to live cells in vitro by locking NDs on the plasma membranes of live cells while inhibiting the endocytic process. We added the biocompatibility of NDs in the last section of the revised manuscript. See Page 18, lines 296-303.

4. Also please comment how you avoided aggregation of nanodiamond? When aggregated (which is unavoidable and in fact extremely difficult to have ND as monodisperse solution without aggregates) sizes are not well controlled, what was the size of your aggregates?

Response:

Figure 3. Dynamic light scattering (DLS) of the NDs showing ~150 nm hydrodynamic diameter.

The description of the preparation of NDs has been added to the Method section of the revised manuscript. See Page 20, lines 350-356.

DLS measurement of the ND solution suggests an average hydrodynamic diameter of 150 nm (Figure 3). We have also checked the sample using TEM (Figure 4), when reasonably dispersed NDs are found. The diameters of NDs are in the range of a few tens of nanometers to ~200 nm, being consistent with the DLS measurement. Although aggregation cannot be completely avoided, it is not significant in the present study (Figure 4). The solution concentration and the

sonication process are important in producing ND solution with satisfactory dispersity. In fact, aggregation of NDs will become significant when smaller sized NDs (e.g., <40 nm) are employed. In such a case, surface treatment/functionalization of NDs will be important to achieve high dispersity [Ivan Rehor, *et.al.*, *ChemPlusChem*. 2014, 79, 21].

Possibility of ND aggregation has been clarified in the revised manuscript and revised supplementary information (See Page 20, lines 351-356 in manuscript, and Page 14, Note 4, Fig. S6 in SI).

Figure 4. A typical TEM image of NDs.

5. And how robustly it was attached to the surface?

Response:

We have checked this by both AFM topography and orientation measurement of NDs before and after the indentation experiments. Neither the ND position nor its orientation is found to change (within the measurement error). These results can be found in the supplementary information, See Page 20, Note 4, Fig. S11.

6. Concluding parts of intro must be much better and showing how we can actually use this method (how easy/difficult, time consuming it is?) and why we need this.

Response:

In the present work, we have adopted long acquisition time during the measurement (for each ODMR spectrum, we have used accumulation time of 1000 s, and 300 s for PDMS, and gelatin experiment, respectively). Therefore, the ~ 5 nm z-direction precision of the deformation measurement is no longer shot-noise limited, but dominated by the environmental fluctuations, such as system mechanical vibration and temperature fluctuation etc.

With the present setting, we are able to study the non-local mechanical property of soft material/biological system in a static manner (when long acquisition time is allowed) with high precision in the direction of deformation (z-direction). Unlike many established indentation-based deformation measurements, which only provide localized deformation profile of the sample, information obtained from the non-local response with high precision and lateral resolution provides additional knowledge that can be used to validate the specific measurement model in determining the mechanical property of homogeneous materials. Moreover, the non-local deformation measurement would potentially enable the study of impact transfer in non-uniform medium at nanometer scale, so that unique mechanical properties of nanocomposite materials with multiple phases and interfaces can be understood. It also adds to the existing biological toolbox a function that can measure the non-local response of live systems upon a local stimulation.

On the other hand, the acquisition time used in the present work is too long to capture the dynamics of many evolving systems, especially the biological ones. Nevertheless, there is space for improvement. In a typical measurement (photon counts $\sim 3.5 \text{ M s}^{-1}$, and ODMR contrast of $\sim 0.8\%$) carried out in the present experiments, the shot-noise-limited sensitivities of the

measured rotation angle is $\sim 1.1^\circ \text{ Hz}^{-1/2}$. This can be achieved by using 1s acquisition time in the ODMR measurement per point of location with the present experimental scheme.

Future application of the wide field ODMR technique can achieve simultaneous, multi-point measurement of deformation with 1s acquisition time.

We have elaborated the significance of the present method in the discussion section of the revised manuscript, see Page 18, lines 296-303.

We have also added the angle sensitivity estimation in the revised supplementary information, see Page 30, Note 9, lines 430-442.

7. I do not agree with ‘potentially dynamical behaviours of systems under mechanical stimuli’; how do you want to achieve this - please comment and give examples.

Response:

The current time resolution (1s, see estimation above) is insufficient to study many dynamic behaviors of live systems, but may still be used for single site investigation of physiological processes in live systems. Furthermore, the application of the wide-field ODMR technique can enable multi-site measurement. For example, one can simultaneously measure the orientation changes of NDs pre-anchored on the plasma membrane as a function of time after a mechanical stimulus (or even other type of stimuli that can be equipped to the AFM tip), so that the “dynamics” of membrane deformation upon impact loading/any other stimuli can be captured.

8. Result section is exceptionally well presented and written. However, authors use wrong nomenclature in some places; e.g. AFM indentation does not allow to measure Young’s modulus, it is Apparent Young’s modulus!

Response:

We thank the referee for pointing this out. We modified “Young’s modulus” to “apparent Young’s modulus” in the revised manuscript and revised supplementary information (List of Changes, Items 4(f) and 8(b)).

9. Another question would be how did you make sure that the tip did not penetrate to the gel/polymers?

There is a likelihood that it could have happened and perhaps selection of colloidal probe could be recommended, did you consider that?

Response:

The smoothness and reversibility of depth-loading curves and the AFM tomographic images after the indentation measurements indicate that most likely the AFM tip penetration did not occur in our experiments.

The proposal of using colloidal probe makes sense, since sharp AFM tips might cause overestimation of the apparent Young's modulus due to excessively overstressed material in the sample [M. Dokukin *et. al.*, *Langmuir* 2012]. We will adopt this suggestion in our future experiments to minimize the tip effects on the samples.

10. You did not show indentation curves to be able to see that so I cannot comment- could you please add them to supplementary file?

Response:

The typical indentation curves are shown in the revised supplementary information (Page 13, Fig. S5; Page 21, Fig. S12; Page 24, Fig. S14; and Page 29, Fig. S18).

11. The other aspect I wanted to ask you to comment on is that both PDMS and Gelatin are homogenous only in macroscale but not on micro and definitely not on nanoscale, how do you account for the fact that chains of both polymers are oriented and entangled in random directions and this may impact on the local deformation - also potentially movements of NDs during deformation. Can you in fact detect this using your method? This would be what we need in testing biological systems. Please comment.

Response:

We have investigated the uniformity of the samples using both point-by-point AFM and the ND sensors, and both methods suggest that the samples used in our experiments were homogeneous down to hundred nanometer scale (see revised supplementary information Note 4, Figs. S5 and S11). Neither rolling nor gliding of NDs on the surface was observed, either in our AFM or orientation measurement before and after the indentation, which suggests that these NDs were attached to the surfaces in a rather robust manner.

On the other hand, we agree that polymers/gel in general can be inhomogeneous at smaller scales (e.g. nanometer scale). As reviewer suggested, the deformation of entangled polymer chains in random directions may result in interesting motion of the NDs (such as non-radial rotation and shift) on the surface. The translational shift might be too small to be resolved in optical imaging, but our orientation measurement provides a unique approach to measuring the complex twisting of polymer network under indentation.

12. Another aspect which is rather minor, why did you use Hertz model? It is not right model in AFM indentation with sharp tip!

Response:

In the theoretical modeling, we used the modified Hertz model (proposed by I. N. Sneddon, 1965. See Ref. 40 in the manuscript) to include the effect of different tip shapes. The tip shape assumed in the simulation was estimated by the SEM image of the used tip (see Reviewer 1 question 3). We revised the terms “Hertzian model” to “Hertz-Sneddon model” in the revised manuscript and supplementary information for clarification (List of Changes, Item 4(b) and Item 8(a)).

13. Methods; I cannot see the method for AFM indentation, we need to see which tip you used (spring constant) this is critical part to say whether you actually could indent to depth that you claim.

Response:

We added methods for AFM indentation in the revised manuscript (Page 19, lines 328-348) and supplementary information (Note 4 & 7, Page 14, lines 234-239 and Page 24, lines 367-373).

14. As previously mentioned AFM does not allow to measure depth of the indentation but it allows to estimate it if we have well-defined experimental setup and modelling protocol in place. Please comment on that.

Response:

We calibrated the AFM setup by performing indentation measurement on sapphire and thermal tune to extract the tip deflection sensitivity and the spring constant, respectively. The tip snap-in point was taken as the contact point and the cantilever deflection was subtracted using pre-calibrated cantilever parameters. The tips we used were made of silicon nitride, whose Young's modulus (in the GPa range) is much larger than those of our samples (in the range of KPa to MPa). As a result, the tip deformation is negligible as compared with the sample deformation. With the contact point set as the snap-in point and the tip deformation taken as zero, we estimated the depth of indentations as the distance between the contact points and the force set points. The methods of estimating the depth of the indentation can be found in the Methods section of the revised manuscript. See Page 19, lines 329-343.

15. Overall, exceptionally good science with outstanding results, which were presented well. Small parts need better clarification; which ND type/size was used, how did you ensure individual ND are attached to the samples, if aggregates how this could influence your results since they are oriented in direct directions (is this a problem- please clarify and comment).

Response:

The NDs used in the present study have an average size of ~150 nm (hydrodynamic diameter measured by DLS, see Figures 3&4 above). On average, each ND contains ~ 500 NV centers. Attachment of NDs on the sample surface is robust, as explained earlier (question 5). Aggregation of NDs is not significant in the present experimental setting (explained earlier, see question 4). Nonetheless, aggregation cannot be completely avoided. In most cases single NDs

were measured, where the 4 pairs of ODMR dips from the NV centers along the 4 crystallographic orientations were observed in the presence of magnetic field. In the presence of aggregation, resonances of NV centers in multiple NDs would be simultaneously recorded, which would result in lower contrasts of the ODMR dips and larger errors in spectral fitting, reducing the precision of the deformation measurement. In our experiments, we only used single NDs. These points have been clarified in the Methods section of the revised manuscript (Page 20, lines 350-356) and revised supplementary information (Page 14, Note 4, Fig. S6).

16. AFM and ND preparation sections are missing.

Response:

The respective sections have been added in the methods section of the revised manuscript, see Page 19, lines 328-356.

17. This manuscript needs much better justification why we need this results/science, and what is the significance of this work. How we can benefit from it? Who need that?

Response:

This has been addressed in earlier questions (Reviewer 1 question 4, and Reviewer 2 question 6), and elaborated in the revised manuscript. See Page 2, lines 29-33; Page 12, lines 202-213; Page 16, lines 257-272; and Page18, lines 284-303.

Reviewer #3 (Remarks to the Author):

"Nanometer-precision non-local deformation reconstruction using nanodiamond sensing", by Xia et al.

The authors demonstrate measurements of surface deformation with the use of the NV colour centre in diamond. The advantage of this technique is shown to rest in the ability to measure these surface deformations at positions away from a probing indenter allowing the mechanical response to be measured in more detail than previously possible. This compliments techniques like nano-indentation where the sensor is the probe.

The concept of the technique is relatively straight-forward to understand and is a unique extension of the utility of the NV centre in diamond. I am of the opinion that this paper warrants publication in Nature Communications. However, the clarity could be greatly improved first as the style and minor grammatical errors detract from work. The sections of the manuscript where the meaning was not clear are discussed below.

Comments:

General:

1. The manuscript describes the new technique quite well using PDMS and gelatin particles as model systems. Of course the focus of the paper is on the implementation of the technique. However, the paper may have broader appeal if a short discussion on the uniqueness of the data relative to other techniques and the new insights it provides is included.

For example, the authors state that the technique reveals the significance of surface/interface effects in material deformation (line 30). Indeed, it is shown that surface tension is required to understand the results. However, the significance and implications of these results are not put into context making it difficult for readers without a the relevant background to appreciate.

Response:

Conventionally the mechanical properties of materials can be studied by measuring the localized deformation profile using indentation based methods (e.g. AFM), but the mechanical property of the material can only be obtained when the correct model is employed, and the choice of model significantly relies on the accurate knowledge of tip parameters and indentation conditions. Unfortunately, the information of tip parameters and indentation conditions is not easily

available, so that one, using local deformations profiles only, would not be able to validate the choice of contact models and consequently determine the mechanical properties of samples. The non-local deformation profile obtained by the proposed method provides additional information that can be used to validate the specific contact models. One such example can be found in the reply to Reviewer 1, question 5. Revision has been made in the manuscript (Page 12, lines 202-213; Page 16, lines 262-272) and revised supplementary information (Page 22 Note 6, lines 326-353) to explain this potential application.

Another example is the identification of surface/interface tension in gelatin experiment, where differences between models with and without surface/interface tension only appeared in non-local deformation profiles of the samples but not in the local profiles. In fact, recent studies [Ref. 16-19] on elastocapillary phenomena suggest that a new model is needed to interpret indentation measurements on soft materials such as cells. For cell measurements carried out in aqueous environment, surface/interface tension could play a significant role, but has not yet been considered in interpreting their mechanical properties. The correct modeling is important to access the stiffness of cells for differentiating cancerous ones from normal ones [K. D. Costa, *Dis. Markers* 2003, 19, 139]. Before we could apply our technique to live cellular systems, here we used gelatin as a model system, since it is a soft matter with Young's modulus (a few kPa) comparable to those of plasma membranes. The measurement of gelatin was carried out in water, an environment similar to that for cell measurements. Using the gelatin as the model sample, we provide the first set of experimental data obtained in aqueous environment and demonstrate the role of surface tension in determining its mechanical properties. This experiment paves the way for investigating mechanical properties/mechano-stimuli induced response of live systems in future.

We added the explanations in the revised manuscript. See Page 16, lines 262-272.

2. Although it is a technical consideration it may be beneficial to include an indication of how long it takes to make this measurement. This might be important for some more dynamic systems as mentioned on line 86 ("dynamical behavior of systems under mechanical stimuli.").

Response:

The issue of dynamic measurement has been elaborated in replies to Reviewer 2 question 6 & 7.

The changes can be found in the revised manuscript (Page 18, Line 299-301) and revised supplementary information (Page 30, Note 9, line 430-442).

3. The Hertzian model appears to describe the indenter force-depth traces in Fig. S12 and Fig. S17 well. Please clarify why the same model is not adequate to describe the data in Fig. 3 and 4 of the main text.

Response:

The difference is caused by the inadequate information obtained from the localized deformation profile. In other words, more than one models fit the data when using only localized deformation profile, but then they are differentiated by the additional information of non-local deformation response. Using gelatin as an example, we see that models with and without the effect of surface tension both well fit the local deformation profiles (Figure S18 in the revised SI). On the contrary, only the model with the surface tension effect fits the non-local deformation profile (Figure 4 in the revised manuscript). Another example can also be found in the reply to Review 1 question 5.

4. Typographical errors:

Line 20:

Acronyms are not defined in the abstract.

Response:

We modified them in the revised manuscript “Atomic force microscopy” and “Nanodiamond” . See Page 2, line 27.

5. Line 28:

"This approach features a 5 nm precision in the loading direction and a sub-hundred nanometer lateral resolution." Does this simply mean 5 nm precision supplied by the AFM normal to the

surface whereas the sub-hundred nano-meter resolution is the diffraction limit and/or the size of the nano-diamond determined by the AFM scan?

Response:

The 5 nm is the precision of the deformation measurement along the indentation direction, as determined by the precision of rotation angle measurement ($\sim 0.5^\circ$, see SI Note 4). The lateral resolution in the single ND measurement is limited by the AFM tip size and the scanning step size. We modified the statement in the revised manuscript. See Page 4, lines 78-82; Page 17, lines 284-287.

6. With the multi-ND approach the resolution is also determined by the ND density which should also be greater than the diffraction limit to avoid convoluted spectra. Could this be concisely clarified further in the text.

Response:

In the multi-ND approach the resolution is currently determined by the ND density and is ultimately limited by the optical resolution to avoid convoluted ODMR spectra. We modified the statement in the revised manuscript. See Page 17, lines 290-295.

7. Line 51:

"Deformation of such large scales may miss the deformation characteristics at finer scale,..."

Suggested rewrite:

"Such techniques lack the sensitivity required to probe important deformation phenomena such as the effect of surface tension at the interface between soft materials and liquids.

Response:

We modified the statement in the revised manuscript according to the referee's suggestion. See Page 3, lines 55-57.

8. Line 64:

"When a nanodiamond is docked on a material surface,..."I recommend replacing the term "dock" with an alternative. The various common usages of this word may cause confusion. For example, this sentence may read: Changes in the rotation of a ND placed on a material surface is directly related to changes in the surface gradient and hence its deformation. They may then be referred to as "surface NDs" to be succinct.

Response

We modified the sentence in the revised manuscript. See Page 4, lines 69-70.

9. Line 70:

Use "ND" or "nanodiamond" consistently.

Response:

We modified the nanodiamond to ND in the revised manuscript and supplementary information (List of Changes, Items 3(d) and (e)).

10. Line 71:

.."multiple NDs docked at various positions..."

"multiple NDs located at various positions".

Response:

We changed the word "dock" to "locate" (List of Changes, Item 3(f)).

11. Line 85:

"which features a precision of < 10 nm,..."

What does this value refer to? deformations provided by the indenter? Or the accuracy in determining the surface gradient (as described on line 179)?

Response:

This value refers to the standard deviation of the reconstructed z value ($\sigma_z = 5.2$ nm, as shown in Page 19, Note 4, Fig. S10 on the revised supplementary information) which defines the precision of deformation measurement.

12. Line 92:

"NDs are docked on the material surface."

NDs are distributed across the material surface.

Response:

We modified the sentence in the revised manuscript. See Page 5, line 98.

13. Line 183:

"The rotation and deformation data are well consistent with the numerical simulation based on a linear elastic bi-layer model."

Consider rewriting this:

"The rotation and deformation data are best described with linear elastic bi-layer numerical simulations."

Response:

We rewrite the sentence according to the referee's suggestion. See Page 11, lines 189-190.

REVIEWERS' COMMENTS:

Reviewer #1 (Remarks to the Author):

I appreciate how the authors addressed the points I raised. Although from my point of view the "appealing" of the manuscript could be still improved, the manuscript can be accepted as it is.

Daniele Passeri

Reviewer #2 (Remarks to the Author):

All comments were addressed well and I am fully satisfied with the revised version of the manuscript.

Reviewer #3 (Remarks to the Author):

The authors have submitted a revised manuscript. The issues raised by myself as referee have been carefully addressed and a detailed response is supplied. In particular, the importance and breadth of the work has more clearly been stated. Technical details related to the limits of the measurement and the models used have also been expanded upon. I believe this greatly enhances the work.

However, the English of the newly appended parts of the manuscript should be checked.

Apart from this, I am satisfied that the work is sound and suitable for publication in Nature Communications.

List of Changes:

We have modified the English in the revised manuscript and revised supplementary information.

Reviewer #3 (Remarks to the Author):

The authors have submitted a revised manuscript. The issues raised by myself as referee have been carefully addressed and a detailed response is supplied. In particular, the importance and breadth of the work has more clearly been stated. Technical details related to the limits of the measurement and the models used have also been expanded upon. I believe this greatly enhances the work.

However, the English of the newly appended parts of the manuscript should be checked.

Response:

We have modified the English in the revised manuscript and revised supplementary information.

Apart from this, I am satisfied that the work is sound and suitable for publication in Nature Communications.